# National and Regional Rates of Chronic Diseases and All-Cause Mortality in Saudi Arabia—Analysis of the 2018 Household Health Survey Data

**DOI:** 10.3390/ijerph20075254

**Published:** 2023-03-24

**Authors:** Majed S. Alzahrani, Yaser S. Alharthi, Jamal K. Aljamal, Abdulrahman A. Alarfaj, Vishal Vennu, Mohammed D. Noweir

**Affiliations:** 1Department of Preventive Medicine, Second Health Cluster, Ministry of Health, Riyadh 12231, Saudi Arabia; 2Department of Pediatric Emergency Medicine, Maternity & Children Hospital, Ministry of Health, Makkah 24269, Saudi Arabia; 3Department of Family Medicine, King Abdulaziz Hospital, Ministry of National Guard, Al-Ahsa 31982, Saudi Arabia; 4Department of Rehabilitation Sciences, College of Applied Medical Sciences, King Saud University, Riyadh 10219, Saudi Arabia; 5Department of Preventive Medicine, Prince Mansour Military Hospital, Taif 26526, Saudi Arabia

**Keywords:** diabetes mellitus, hypertension, cardiovascular diseases, cancer, mortality, Saudi Arabia

## Abstract

The disease burden and mortality were estimated in Saudi Arabia between 2010 and 2017 but were unknown in 2018. This study aims to assess the 2018 national and regional rates of chronic diseases and all-cause mortality among the total and Saudi populations. In this descriptive cross-sectional study, we obtained data from 24,012 households from the 2018 household health survey. We included doctor-diagnosed chronic conditions such as diabetes mellitus (DM), hypertension (HTN), cardiovascular diseases (CAD), and cancer (CN). A secondary analysis was performed by the total and Saudi populations. Both citizens and residents comprised the total population. Makkah and Al-Medina had greater rates among the total population; however, Al-Baha and Ha’il had high rates of chronic diseases and related mortality in the Saudi population. Age-adjusted mortality rates were 286 per 100,000 population-year. The age-adjusted mortality rate among those aged 65 and above was 3428 per 100,000 population in the same age group. Men had a rate of 1779 per 100,000 men, which was higher than the rate of 1649 for women. In 2018, most citizens in Ha’il had DM, most Al-Baha had HTN and CAD, and most Al-Qassim had CN. People aged 65 and older had the highest death rate.

## 1. Introduction

The word “chronic” is often used for a disease lasting over three months. The phrase “chronic disease” represents great diversity in the academic literature and professional communities. Non-communicable diseases are characterized by their non-infectious origins and are frequently linked to chronic disorders [1]. Globally, the most common causes of death and disability are chronic diseases, which are silent pandemics [2]. From 27 million in 1990 to 39.5 million in 2016, deaths due to chronic diseases increasingly engulf the entire world’s population. Demographic shifts are accompanying the global epidemic of chronic diseases. The World Health Organization (WHO) estimates that chronic diseases account for 38 million (63%) annual fatalities [3]. In the United States, at least two chronic illnesses affect 40% of adults [3,4].

Globally, illnesses in newborns, ischemic heart disease, stroke, lower respiratory infections, and chronic obstructive pulmonary disease were the top five causes of disability-adjusted life years in 2017. Global disability-adjusted life years for several chronic diseases rose between 1990 and 2017, despite declining age-standardized rates [5]. Moreover, worldwide life expectancy at age 70 has increased, largely due to declines in chronic diseases [6]. According to data from the 2019 Global Burden of Diseases, Injuries, and Risk Factors Study [5], ischemic heart disease, stroke, chronic obstructive pulmonary disease, Alzheimer’s disease, other dementias, and lower respiratory infections were the top five level 3 causes of death globally for people aged 70 and over. Alzheimer’s disease and other dementias, lung cancer, diabetes, and chronic kidney disease all saw increased death rates among those aged 70 and older between 1990 and 2019.

According to a Global Burden of Disease country-specific report for Saudi Arabia, chronic diseases were the primary cause of death in 2010 [7]. This report has demonstrated that increased body mass index (BMI) was the leading risk factor for disease in men and women (7000 and 4610 per 100,000 people, respectively). High glucose levels were the second-leading disease risk factor for women (3280 per 100,000 people) and the third-leading risk factor for men (6250 per 100,000 people). The Saudi Health Interview Survey findings showed that obesity, hypertension, and diabetes are severe problems in Saudi society [8]. There were 50,600 cases of diabetes, 51,000 cases of hypertension, 49,000 cases of asthma, and 45,000 cases of obesity per 100,000 persons. A study also reported a high BMI among the Saudi population as a significant risk factor for disease burden [7].

Utilizing information from the 2017 Global Burden of Disease Study, researchers recently determined levels and temporal patterns in mortality, health loss, risk factors, and healthcare services in Saudi Arabia between 1990 and 2017 [9]. However, the prevalence of chronic diseases and all-cause mortality in Saudi Arabia and its 13 administrative regions in 2018 is unknown. These statistics should be widely known to align with Saudi Vision 2030 goals of strengthening and transforming healthcare in Saudi Arabia [10,11]. Consequently, this study aimed to assess the prevalence of chronic diseases and all-cause mortality among Saudi citizens and the total population in 2018.

## 2. Materials and Methods

The 2018 Household Health Survey provided the data for this descriptive cross-sectional investigation. In 2018, the General Authority for Statistics (GaStat) conducted this survey of all Saudi Arabian residents, both Saudis and non-Saudis. Saudis were considered citizens, whereas non-Saudis were referred to as residents. The study visited a random sample unit of households distributed throughout all administrative regions in Saudi Arabia. Data were collected from n = 24,012 households using an electronic questionnaire. 

The GaStat’s statisticians designed the computerized survey. The GaStat provided relevant organizations, professionals, and specialists in household health with this questionnaire to solicit their perspectives and ideas. The questionnaire’s questions were revised utilizing a specific scientific technique to standardize the question forms used by researchers. The GaStat converted the survey form electronically to collect data. The skilled field staff used tabs to collect the data.

All gathered data were checked in the supervision area by the field researcher, their inspector, and the survey supervisor. The classification and coding inputs created during the data-gathering phase were used to disaggregate the raw data. Data specialists from GaStat carried out data processing and analysis based on protocols [12], particularly data anonymization, thus guaranteeing data confidentiality. After the final evaluation by experts using cutting-edge software and technology, data specialists conducted the primary analysis to extract results and upload them to the database.

A specialist doctor identified a chronic condition and performed the required tests to inform the patient of this chronic disease. These analyses included diabetes mellitus (DM), hypertension (HTN), cardiovascular diseases (CAD), and cancer as chronic illnesses (CN). The permanent loss of all signs of life following a live birth, such as breathing, pulse, or voluntary movement, was called “all-cause mortality”. We eliminated those not diagnosed by a specialist and unaware of their disease.

We conducted a secondary analysis of the study’s data of 24,012 households. The analysis was performed on the total and Saudi populations. The total population included Saudis (citizens) and non-Saudis (residents). Saudi citizens were referred to as the Saudi population. We calculate the age-adjusted rates per 100,000 population-year and event rate for each chronic disease by stratifying the total population and Saudi citizens across all 13 administrative regions. Moreover, we calculated the age-adjusted all-cause mortality rate per 100,000 population-year by stratifying total and gender. We created a 95% confidence interval for each rate using the following formulas: lower limit = (100,000/n) (d − (1.96 × square root of d) and upper limit = (100,000/n) (d + (1.96 × square root of d), where d = the rate and n = the denominator of the rate. The total or general population of Saudi Arabia was defined as including Saudis (citizens) and non-Saudis (residents). The denominator was the entire population, subdivided by country, region, age, and gender. We performed all the calculations using Microsoft Excel 2019 (Microsoft Corporation, Redmond, Washington, USA).

## 3. Results

Of the total population, there were 7931 DM, 7009 HTN, and 1168 CAD cases per 100,000 population in Makkah. The total population of 176 per 100,000 population-year in Riyadh had a CN. In Al-Baha, the Saudi population had HTN and CAD at 5451 and 909 per 100,000 population-year, respectively. In Najran, there were low rates of 4699 DM, 3369 HTN, and 385 CAD per 100,000 population-year. In Makkah, there were low rates of 92 CN per 100,000 population-year.

The Saudi population of 5933 per 100,000 population-year in Ha’il had a DM. In Al-Baha, the Saudi population had HTN and CAD at 5451 and 909 per 100,000 population-year, respectively. In Al-Qassim, the Saudi population had CN at a rate of 171 per 100,000 population-year. The Saudi population of Najran had lower rates of DM, HTN, and CAD than other towns. The town of Makkah and Al-Jawf had very fewer rates of CN in the Saudi population (75 per 100,000 population-year) (Table 1).

In Makkah and Ha’il, 11,034 men and 10,104 women had DM per 100,000 population-year, respectively. There were 7607 men and 10,455 women with HTN per 100,000 population-year in Al-Madina. Al-Baha and Al-Madinah had 1758 men and 1736 women with CAD per 100,000 population-year, respectively. Per 100,000 population-year, 254 men in Tabuk and 402 women in Ha’il had CN. In Asir, men experienced DM at a lower rate of 7002 per 100,000 population-year. Nonetheless, HTN and CAD fever affected both men and women in Najran. In Tabuk and Ha’il, respectively, there were 45 women and 50 men with very low rates of CN per 100,000 population-year (Table 2).

Men Saudi citizens in Makkah had DM and HTN rates of 11,516 and 9524 per 100,000 population-year, respectively. A DM rate of 1650 per 100,000 population-year was observed among women Saudi citizens in Al-Jawf. In Makkah, the HTN rates for men and women were 9524 and 10,990 per 100,000 population-year, respectively. The CN rates for men in the Northern borders and women in Ha’il were 350 and 477 per 100,000 population-year, respectively. The DM, HTN, and CAD rates among men citizens in Al-Qassim, Tabuk, and Najran were lower at 8260, 6042, and 864 per population-year, respectively. The DM and CAD of women citizens of Najran were lower (each 650 per 100,000 population-year). In Ha’il and Tabuk, the CN rates for men and women citizens were 82 and 53 per 100,000 population-year, respectively (Table 3).

The national and regional rates per 100,000 population-year for any chronic disease are shown in Figure 1 for both the total population and the Saudi population. Al-Baha, Ha’il, Jizan, and Asir had high rates among the Saudi people, although Makkah, Al-Medina, and Jizan had greater rates among the general population. These four areas all had rates that were higher than the national average. In addition, Jizan had a higher-than-average prevalence of any chronic condition in Saudi and overall populations, although Najran had a lower rate in both. The rate of any chronic disease per 100,000 population-year increased with age groups (Figure 2).

In 2018, there were 286 (95% CI = 275–847) age-adjusted deaths per 100,000 population in Saudi Arabia. The event rate of mortality was 28.6%. As the population aged, the death rate increased. According to the total population of adults aged 65 and above, the age-adjusted all-cause death rate was 3428 per 100,000 among the same age group; men had a rate of 1779 per 100,000 men, which was higher than the rate of 1649 for women (Figure 3).

## 4. Discussion

Makkah’s general populace had greater rates of DM, HTN, and CAD; while most of the Saudi population in Ha’il had DM, the Al-Baha Saudi population had both HTN and CAD. Most of the Al-Qassim Saudi population had CN. Al-Madina women had HTN. In contrast, most men in Makkah had DM in Al-Baha, and CAD rates were higher for both men and women. In Ha’il, women had a higher CN rate. DM was more prevalent among Saudi men in Riyadh. Chronic conditions were more prevalent among the Saudi population in Al-Baha and Ha’il than the national average. No matter the demographic type, DM rates were higher in people over 25. Any chronic condition is more prevalent as people age. People aged 65 years and older had the greatest death rate.

The results of this study are consistent with the literature [9,13,14,15] and results of the Saudi Health Interview Survey, which employs a representative sample of people aged 15 and older and covers all 13 regions of Saudi Arabia [8]. According to this survey, smoking, high blood pressure, high cholesterol, obesity, and diabetes are major problems affecting an increasing number of people in Saudi Arabia. One of the causes may be obesity and body fat distribution, which are linked to many chronic diseases in Saudi Arabia’s population [7]. The link between obesity and numerous chronic diseases may be linked to risk factors such as smoking, eating poorly, and not exercising [16,17,18]. Since healthcare in Saudi Arabia is free and widely available, it is surprising that individuals only go to the doctor when sick and do not take advantage of preventive care [16]. 

According to the data, there were 286 deaths per 100,000 people from all causes in 2018 after accounting for age. According to this result, the mortality rate from all causes has decreased since earlier findings [9]. Saudi Arabia had an age-standardized all-cause death rate of 634 per 100,000 people in 2017, down from 740 per 100,000 in 2010 and 832 per 100,000 in 1990. These figures are consistent with a reduction in age-standardized mortality from 1990 to 2010 at an annualized rate of change of −0.58%, which accelerated from 2010 to 2017 (−2.20%). Moreover, Saudi Arabia’s age-standardized mortality rate declined even further from 2017 to 2018—an annual rate of change of −3.54%—meaning that this study’s conclusions align with the previous research. One explanation for the annual rate of change between 2017 and 2018 could be the creation of numerous unique solutions to address the country’s health issues due to the collaboration between the Saudi Health Ministry and the Institute for Health Metrics and Evaluation [16].

A recent study estimates that there were 6.55 million deaths from 122 million incidents and 101 million common stroke episodes worldwide in 2019 [19]. From 1990 to 2019, the death rates among those over 70 increased globally for lung cancer, diabetes, chronic renal illness, and Alzheimer’s disease by 29.3%, 11.7%, 16.3%, and 31.9%, respectively [6]. The most current study found that, in the Middle East and North Africa region, Parkinson’s disease had an age-standardized point prevalence of 82.6 per 100,000 people and an age-standardized mortality rate of 5.3 in 2019, both of which had increased from 1990 to 2019 by 15.4% and 2.3%, respectively [20]. The fact that preventative programs have not previously targeted these diseases could be one explanation. Interventions targeting illnesses that gradually reduce physical functionality may be required to change this trend. Physical function limitations were the primary causes of disease and all-cause mortality, especially among older adults [21].

The findings from this study have significant implications for encouraging those who choose not to utilize a nation’s free and publicly available universal healthcare, which the United States is currently debating whether or not to offer [16]. These results may help the country comprehend initiatives targeted at collecting data to increase the understanding of the epidemiology of chronic diseases. These American programs aim to collect epidemiological data on chronic diseases and show how this knowledge might help combat chronic diseases [22]. This finding, while preliminary, suggests investigating a relationship between social variables, chronic condition prevalence, and outcomes [16], especially because loneliness has the same negative health impacts as smoking and being overweight, according to some research [23], and was linked to a person having more chronic illnesses [24].

These findings may help the country better understand chronic illnesses’ economic effects. According to a study [25], Americans spent USD 2243 more on average if suffering from a chronic disease. Financial stress may make it harder for people to take their medications as prescribed [26,27]. In several other nations, laws protect those with chronic diseases from bearing too high a financial burden. For example, France lowered copayments for individuals with chronic conditions as of 2008, and Germany restricts cost sharing to 1% of income as opposed to 2% for the entire population [28]. Saudi Arabia is a wealthy country. A large portion of this wealth has been allocated to the Saudi people by providing free healthcare to anybody who can prove their residency legally, whether or not they are Saudi [29].

Implementing public health programs is one of the concerns that arise from these findings. These initiatives significantly raise public awareness of chronic diseases, encourage healthy lifestyles, and educate the public [18]. Most of the program’s implementation has been placed in the hands of local agencies and community-based organizations, even though they may benefit from funding at various levels, such as state, federal, and private [30]. Studies have demonstrated that public health initiatives can lower the death rates of conditions such as cancer, diabetes, and cardiovascular disease. Still, the outcomes can vary depending on the situation and the program type [31]. For example, depending on the type of cancer, the results of various approaches to cancer prevention and screening differed substantially [32]. However, programs may assist in improving outcomes and saving medical costs by encouraging people with chronic diseases to keep up with their outpatient care and attend planned appointments [33].

The development of policies in Saudi Arabia for preventing chronic diseases and the problems linked to them, particularly regarding the country’s obesity problem, will be significantly impacted by this finding. Given that more than half of the world’s overweight population lives in Saudi Arabia, these strategies are essential and might serve as a call to action [34]. Additionally, people who are obese encounter social discrimination and a decline in health-related quality of life, both of which have a detrimental effect on their emotional and mental health [35]. This approach ultimately results in chronic diseases that burden the country. Due to the poor rates of obesity diagnosis, documentation, and care linked to inadequate awareness of the causes of being overweight, strategies are essential. A call to action is necessary regarding obesity in Saudi Arabia [36].

The current study’s findings are that country-specific benchmarks can be used to create and implement health intervention programs to address an overweight population and lessen the burden of chronic disease and disability while tracking regional estimates due to a sharp rise in the number of older adults in the country by 2050 [37]. This research provided cause- and risk-specific information that might be used to develop strategies targeting men, women, and other sociodemographic groups to reduce chronic diseases and all-cause mortality. The relative paucity of information could be an impending problem and emphasizes the urgent need for monitoring systems in areas with little information on risk factors and the prevalence of non-fatal diseases in older populations. We could not examine the impact of COVID-19 on mortality because our analysis covered the year 2018, which was before the COVID-19 epidemic. According to international studies [38,39], most deaths occurred among people over 70, with a case fatality rate of more than 10% and a potential increase of 30%.

This study’s main strength is the large sample size from all 13 administrative regions of Saudi Arabia, including information on non-Saudi expatriates. This makes it possible to apply the results to the entire Saudi population. Other advantages are the methodology and sampling utilized to gather the data [12]. This research has several limitations, however. First, it is unknown whether sick people were exposed to this survey [40]. Second, one cannot rule out the possibility that genetic predisposition contributes to the high incidence of DM and CVD risk factors in the Saudi population, which may be linked to high consanguinity [41]. Third, the shortcomings of currently available analytical tools may not accurately depict temporal changes in Saudi Arabia’s mortality, incidence, and prevalence [9]. Finally, the detection of risk factors and disease burden may have improved over time in a nation such as Saudi Arabia, a continuously developing country, which could cause some of the reported increases in the prevalence of chronic diseases.

## 5. Conclusions

In 2018, Makkah’s populace had greater CAD, HTN, and CN rates. Most of the Saudi population in Ha’il had DM, a majority in Al-Baha had both HTN and CAD, and a majority in Al-Qassim had CN. People aged 65 years and older had the highest death rates. Although the estimates of chronic diseases and all-cause mortality are limited to 2018 in this study, a call to action to adopt wellness policies for Saudi Arabia’s overweight population has been associated with the development of chronic diseases.

## Figures and Tables

**Figure 1 ijerph-20-05254-f001:**
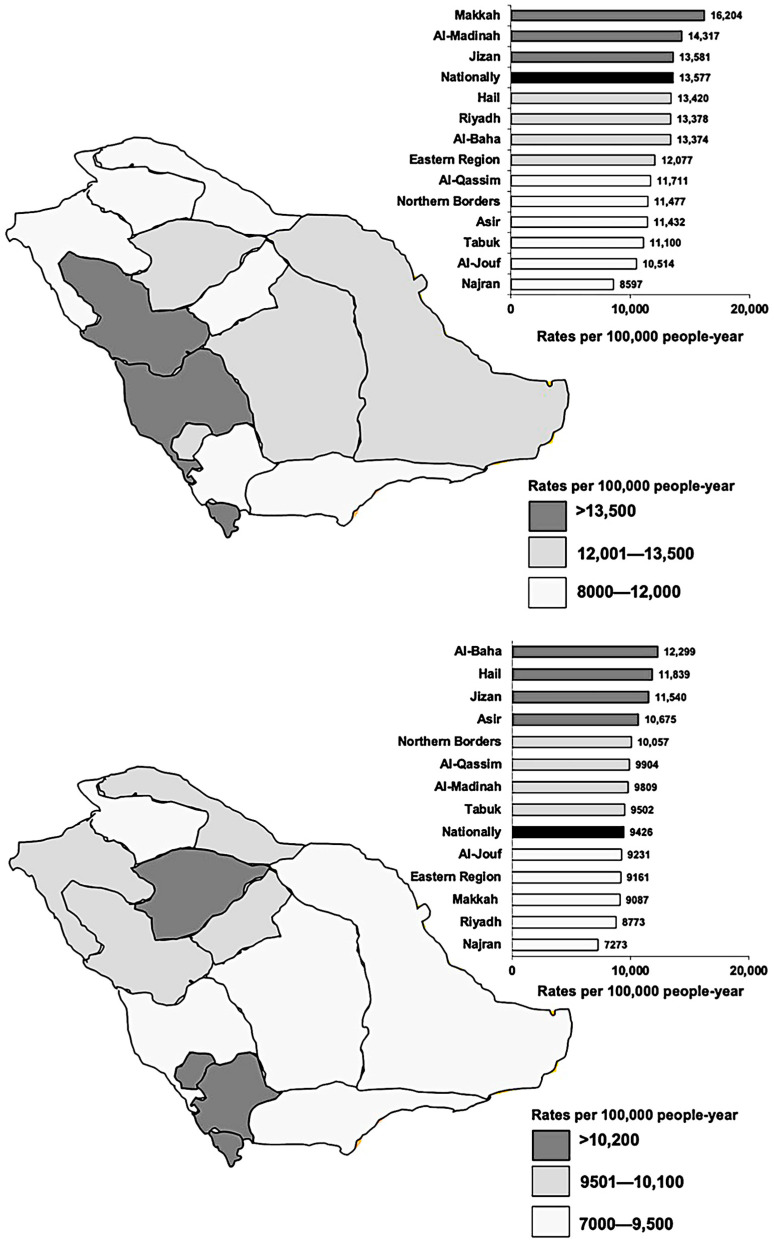
National and regional chronic disease rates by (**upper**) total population and (**lower**) Saudi population.

**Figure 2 ijerph-20-05254-f002:**
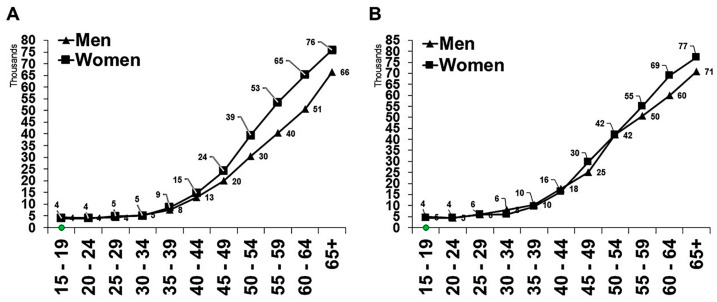
Age-adjusted rates of any chronic disease in (**A**) total population and (**B**) Saudi population by gender. Note. Rates are presented per 100,000 population-year.

**Figure 3 ijerph-20-05254-f003:**
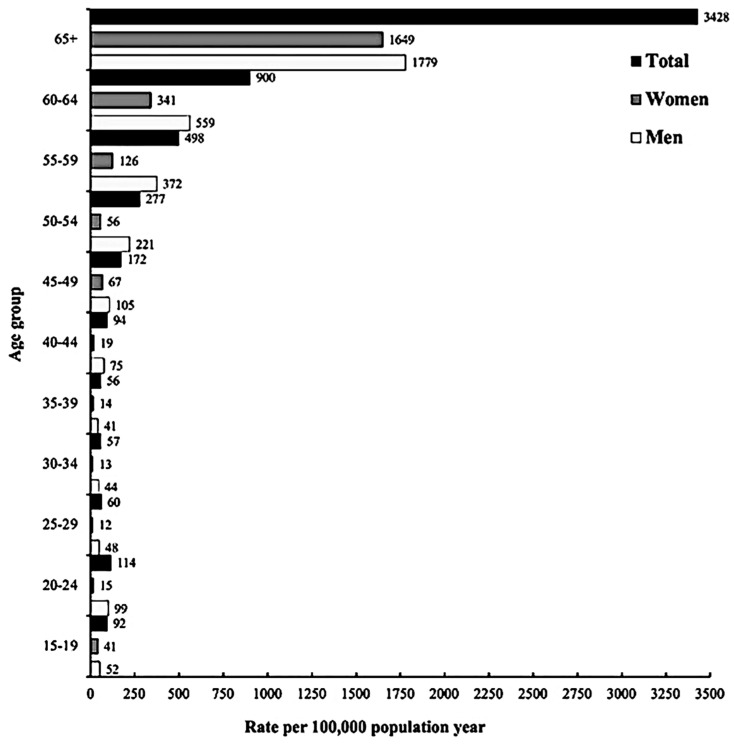
Age-adjusted all-cause mortality rate by total and gender.

**Table 1 ijerph-20-05254-t001:** The rates of chronic diseases per 100,000 population-year in total and Saudi populations by the administrative region.

The Administrative Region	No. of Households	No. of Population	Total Population ^π^				Saudi Population ^ψ^			
			DM Rate (95% CI) (%) ^¶^	HTN Rate (95% CI) (%) ^¶^	CAD Rate (95% CI) (%) ^¶^	CN Rate (95% CI) (%) ^¶^	DM Rate (95% CI) (%) ^¶^	HTN Rate (95% CI) (%) ^¶^	CAD Rate (95% CI) (%) ^¶^	CN Rate (95% CI) (%) ^¶^
Riyadh	3726	8,216,284	6494 (6234–19,222) (6.5)	5844 (5610–17,297) (5.8)	864 (830–2558) (0.9)	176 (169–520) (0.2)	4176 (4009–12,360) (4.2)	3777 (3626–11,180) (3.8)	689 (662–2040) (0.7)	131 (125–387) (0.1)
Makkah	4356	8,557,766	7931 (7614–23,476) (7.9)	7009 (6729–20,746) (7.0)	1168 (1122–3458) (1.2)	96 (92–283) (0.1)	4266 (4096–12,628) (4.3)	3987 (3828–12,360) (4.0)	747 (717–2211) (0.7)	87 (83–256) (0.1)
Al-Madinah	1656	2,132,679	6433 (6176–19,042) (6.4)	6704 (6436–19,843) (6.7)	1054 (1012–3121) (1.1)	126 (121–372) (0.1)	4696 (4508–13,899) (4.7)	4292 (4120–12,703) (4.3)	696 (669–2061) (0.7)	126 (121–372) (0.1)
Al-Qassim	1188	1,423,935	5670 (5443–16,783) (5.7)	4999 (4799–14,796) (5.0)	870 (836–2577) (0.9)	171 (165–508) (0.2)	4527 (4346–13,399) (4.5)	4421 (4244–13,085) (4.4)	785 (754–2324) (0.8)	171 (165–508) (0.2)
Eastern Region	3024	4,900,325	5936 (5699–17,571) (5.9)	5209 (5000–15,418) (5.2)	794 (762–2351) (0.8)	138 (133–409) (0.1)	4303 (4131–12,736) (4.3)	4048 (3886–11,983) (4.0)	671 (644–1987) (0.7)	138 (133–409) (0.1)
Asir	1620	2,211,875	5354 (5140–15,848) (5.4)	4993 (4794–14,780) (5.0)	944 (906–2794) (0.9)	141 (135–417) (0.1)	4970 (4772–14,713) (5.0)	4667 (4480–13,813) (4.7)	897 (861–2654) (0.9)	141 (135–417) (0.1)
Tabuk	1314	910,030	5933 (5696–17,562) (5.9)	4302 (4130–12,734) (4.3)	743 (713–2199) (0.7)	122 (117–362) (0.1)	4928 (4731–14,588) (4.9)	3791 (3640–11,223) (3.8)	702 (674–2077) (0.7)	81 (78–239) (0.1)
Ha’il	1170	699,774	7072 (6789–20,933) (7.1)	5312 (5100–15,724) (5.3)	880 (845–2606) (0.9)	155 (149–459) (0.2)	5933 (5695–17,561) (5.9)	4911 (4714–14,536) (4.9)	840 (806–2486) (0.8)	155 (149–459) (0.2)
Northern borders	1260	365,231	6231 (5982–18,444) (6.2)	4215 (4046–12,476) (4.2)	877 (842–2596) (0.9)	154 (148–456) (0.2)	5,13 (4926–15,188) (5.1)	3995 (3835–11,824) (4.0)	795 (763–2352) (0.8)	137 (132–406) (0.1)
Jizan	1278	1,567,547	6817 (6544–20,178) (6.8)	5822 (5589–17,232) (5.8)	784 (752–2319) (0.8)	160 (153–473) (0.2)	5744 (5514–17,003) (5.7)	4962 (4763–14,687) (5.0)	743 (713–2198) (0.7)	91 (88–270) (0.1)
Najran	1152	582,243	4699 (4511–13,909) (4.7)	3369 (3235–9973) (3.4)	385 (370–1140) (0.4)	143 (138–424) (0.1)	3733 (3583–11,049) (3.7)	3037 (2915–8989) (3.0)	385 (370–1140) (0.4)	119 (114–351) (0.1)
Al-Baha	1080	476,172	6264 (6013–18,541) (6.3)	5940 (5702–17,582) (5.9)	1056 (1014–3127) (1.1)	113 (109–336) (0.1)	5825 (5592–17,243) (5.8)	5451 (5233–16,136) (5.5)	909 (873–2691) (0.9)	113 (109–336) (0.1)
Al-Jawf	1188	508,475	5202 (4994–15,398) (5.2)	4270 (4099–12,638) (4.3)	882 (847–2611) (0.9)	161 (154–476) (0.2)	4468 (4290–13,227) (4.5)	3874 (3719–11,466) (3.9)	814 (782–2411) (0.8)	75 (72–222) (0.1)
Total	24,012	32,552,336	6624 (6359–19,607) (6.6)	5875 (5640–17,389) (5.9)	940 (902–2781) (0.9)	138 (133–410) (0.1)	4488 (4308–13,284) (4.5)	4093 (3929–12,114) (4.1)	727 (698–2153) (0.7)	118 (114–350) (0.1)

Abbreviations: CI, confidence interval; DM, diabetes mellitus; HTN, hypertension; CAD, cardiovascular diseases; CN, cancer. Note. Rates are presented per 100,000 population-year. ^π^ The total population of Saudi Arabia was defined as including Saudis (citizens) and non-Saudis (residents). ^ψ^ Saudi citizens were referred to as the Saudi population. ^¶^ Event rate.

**Table 2 ijerph-20-05254-t002:** The rates of chronic diseases in the total population by gender and the administrative region.

The Administrative Region	Men				Women			
	DM	HTN	CAD	CN	DM	HTN	CAD	CN
Riyadh	8522	7274	1244	156	7945	7791	878	336
Makkah	11,034	8665	1595	75	8589	9157	1309	187
Al-Madinah	8169	7607	1131	116	8815	10,455	1736	233
Al-Qassim	6832	5354	1217	120	8058	8115	982	374
Eastern Region	7077	5891	844	88	8401	7905	1296	324
Asir	7002	5318	1272	64	7250	8305	1231	343
Tabuk	8111	5095	1202	254	8146	7010	759	45
Ha’il	8519	6248	1150	50	10,104	7788	1140	402
Northern borders	7857	4904	1420	226	9158	6732	878	185
Jizan	9386	6573	1183	187	8680	9201	868	244
Najran	7190	4407	523	55	5399	4890	532	388
Al-Baha	8156	6154	1758	80	8046	9533	896	228
Al-Jawf	6488	4773	1135	104	8171	7528	1334	396
Total	8730	7008	1256	113	8265	8422	1145	276

Abbreviations: DM, diabetes mellitus; HTN, hypertension; CAD, cardiovascular diseases; CN, cancer. Note. Rates are presented per 100,000 population-year.

**Table 3 ijerph-20-05254-t003:** The rates of chronic diseases in the Saudi population by gender and the administrative region.

The Administrative Region	Men				Women			
	DM	HTN	CAD	CN	DM	HTN	CAD	CN
Riyadh	10,530	8905	2127	280	1314	10,200	1314	383
Makkah	11,516	9524	2212	132	1616	10,990	1616	316
Al-Madinah	10,883	8781	1752	230	1359	10,413	1359	332
Al-Qassim	8260	8164	1901	222	1230	9548	1230	469
Eastern Region	9074	7990	1342	183	1632	9953	1632	438
Asir	9626	7410	1793	100	1421	9222	1421	396
Tabuk	9285	6042	1686	240	881	8033	881	53
Ha’il	10,128	9070	1717	82	1354	8849	1354	477
Northern borders	8807	6876	1893	350	1037	7844	1037	156
Jizan	11,159	7867	1677	60	983	9944	983	268
Najran	8408	6343	864	91	650	5598	650	376
Al-Baha	10,858	8055	2151	121	1022	10,485	1022	260
Al-Jawf	8834	6798	1724	183	1650	9312	1650	127
Total	10,261	8451	1873	183	1388	9977	1388	352

Abbreviations: DM, diabetes mellitus; HTN, hypertension; CAD, cardiovascular diseases; CN, cancer. Note. Rates are presented per 100,000 population-year.

## Data Availability

The data supporting this study’s findings are openly available in the General Authority for Statistics at https://www.stats.gov.sa/en/965 (accessed on 9 March 2023).

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
