# Peer review of "National and Regional Rates of Chronic Diseases and All-Cause Mortality in Saudi Arabia—Analysis of the 2018 Household Health Survey Data"

_ijerph, 2023, doi:10.3390/ijerph20075254_

Round 1
Reviewer 1 Report (Previous Reviewer 3)
The authors looked at national and regional rates of chronic diseases and all-cause mortality in Saudi Arabia. I found many errors after a detailed review.
1. Mortality rate was calculated incorrectly. Authors reported " Age-adjusted mortality rates were 286 per 100,000 population-year; 27 men had 177 per 100,000 men than women (109 per 100,000 women)." It is apparent that there were 286 deaths per 100,000 population, and among those who died, 177 were men and 109 were women. However, the reported mortality rate for men and women are simply wrong.
2. I have found numerous inconsistent reporting in the results text vs. what's presented in table 2 and table 3.
3. Figure 3 is hard to read, should improve presentation.
4. I have seen a few spellings error, e.g. the administrative regions.
5. Formulas are un-necessary.
6. Consider report confidence intervals along with event rates, at least for main results.
Author Response
Reviewer #1
The authors looked at national and regional rates of chronic diseases and all-cause mortality in Saudi Arabia. I found many errors after a detailed review.
Response: We are grateful for the reviewers' comments on the current condition of our paper. Every inaccuracy has been corrected and amended.
- The mortality rate was calculated incorrectly. Authors reported, "Age-adjusted mortality rates were 286 per 100,000 population-year; 27 men had 177 per 100,000 men than women (109 per 100,000 women)." It is apparent that there were 286 deaths per 100,000 population, and among those who died, 177 were men, and 109 were women. However, the reported mortality rate for men and women is wrong.
Response: Thank you, As we understand, the mortality rate among men and women was equal to total mortality (177 in men + 109 in women = 286 in total). However, the reported mortality for men and women was deleted by agreeing with the reviewer in the abstract on page #1, line 30, and in the main text on page #7, line 189.
- I have found inconsistent reporting in the results text vs. what's presented in table 2 and table 3.
Response: Thank you. The inconsistent reporting of results in the text compared to what was presented in Tables 2 and 3 has been addressed and revised accordingly on pages 4 and 5, lines 141-164.
- Figure 3 needs to be easier to read. It should improve presentation.
Response: Thank you. As suggested, Figure 3 has been improved presentation to be read easily on page #7, line 194.
- I have seen a few spellings error, e.g. the administrative regions.
Response: Thank you, and all spelling errors, including the administrative region, have been addressed throughout the paper.
- Formulas are unnecessary.
Response: Thank you. As suggested, the formulas are deleted on page #3, line 118.
- Consider reporting confidence intervals and event rates for the main results.
Response: Thank you. As suggested, the main results (chronic diseases and overall all-cause mortality) are updated with confidence intervals and event rates (see Table 1 on pages #3 and 4).

Reviewer 2 Report (Previous Reviewer 1)
Thank you for making necessary changes.
Author Response
Reviewer #2
Thank you for making the necessary changes.
Response: After responding to the reviewer's helpful suggestions, we'd want to thank them for being satisfied with the paper as it stands right now.

Round 2
Reviewer 1 Report (Previous Reviewer 3)
Thanks for the authors to revise the manuscript considerably. The manuscript was improved, but I do see a few more errors.
1. Line 139: "Al-Baha had 1,758 men and 1,736 women with CAD per 100,000 139 population year. Per 100,000 population year, 254 men and 402 women in Ha'il had CN." This statement is different from table 2.
2. Note there is an error in Table 3 "The administrative re1.9gion".
3. In the text "Northern region" was mentioned, but in table it was listed as "Northern borders". Should make these consistent.
Author Response
Reviewer #1
Thanks for the authors revising the manuscript considerably. The manuscript was improved, but I do see a few more things that could be corrected.
Response: we would like to thank the reviewer for this positive feedback on our revisions and manuscript state. As suggested, a few more errors have been taken care of.
- Line 139: "Al-Baha had 1,758 men and 1,736women with CAD per 100,000 139 population year. Per 100,000 population year, 254 men and 402 women in Ha'il had CN." This statement is different from table 2.
Response: Thank you to the reviewer for spotting these valuable errors. All these errors have been corrected on page 4, lines 139-141.
- Note an error in Table 3, "The administrative 9gion".
Response: Thank you, the error has been corrected in Table 3 on page 5.
- In the text, "Northern region" was mentioned, but in the table, it was listed as "Northern borders." We should make these consistent.
Response: Thank you, and as you said, for consistency, we have changed it to Northern borders in the text on page 5, line 155.

This manuscript is a resubmission of an earlier submission. The following is a list of the peer review reports and author responses from that submission.
Round 1
Reviewer 1 Report
Comments have been added to the attached draft.

Author Response
Reviewer #1
Abstract
Define Saudi vs. non-Saudi households.
Response:
Define incredible death rate.
Response: As suggested, it has been defined on page 1, lines 25-26.
Introduction
What do you mean by 38 M out of 63%? Rewrite it for easy understanding.
Response: As suggested, it has been rewritten on page 1, line 48.
Page 2, line 50: 50,600, 51000; Confusing to state two numbers like these. It is like that throughout this draft. Please make the necessary changes to make sure everything is clear.
Response: As suggested, the necessary changes have been made throughout the draft, including on page 2, lines 69-70.
Page 2, line 59: Please explain Saudi vs. the general population.
Response: As suggested, it has been explained on page 3, lines 109-111.
Methods
Use a different name for different definitions. You cannot use the same rate for all different purposes.
Response: As suggested, the different names for rate have been used on page 3, lines 115-131.
Table 1. How the total population was different from the Saudi population. Please define.
Response: The definition has been provided on page 3, lines 109-111.
Results
Figure 1: Words in this graph and a few others throughout this draft are illegible. Please make the necessary changes so that reader can read them easily.
Response: As suggested, we made the necessary changes to Figure 1 on page 6.
Page 8, lines 164-65: Is there a significant difference statistically?
Response: We appreciate the reviewer for this important comment. However, the purpose of our study is to present the national and regional rates of statistics.
Discussion
Page 7, lines 170-173: The higher rate among one gender or region has been stated, but no statistical analysis was performed to support that claim. And it is the same throughout this draft.
Response: We appreciate the reviewer for this important comment. However, the purpose of our study is to present the national and regional rates of statistics.
Page 7, lines 190-192: Can you explain how 286 deaths in 2018 are consistent with 634 death in 2017 and so on?
Response: Thank you to the reviewer for this vital comment. The sentence has been modified on page 8, lines 231-234.
Page 8, line 197: So, it differs between 2017 and 2018 from previous years?
Response: Yes, and it has been noted.
Page 8, lines 235-36: Can you please verify this claim again, as it used the 2016 reference, which might be outdated now? Thank you.
Response: We appreciate the reviewer for this comment. As suggested, the reference has been claimed with a new recent one.

Reviewer 2 Report
The manuscript reports the national and regional rates of chronic diseases and all-cause mortality in Saudi Arabia depending on health survey data in 2018. The paper is not suitable for publication in this form, it needs revisions. Some of the revisions are listed below:
According to the "Instructions for Authors", the abstract should be a single paragraph and should follow the style of structured abstracts, but without headings. Please remove the headings.
The introduction should be significantly improved. The current state of the research field should be included and related publications should be added. Only general information is given, and international studies about this topic should be reviewed.
Studies about the international status of chronic diseases should be included to discuss the results. The final conclusions should include key findings and be more specific. Please revise and include some key findings.
Author Response
Reviewer #2
The manuscript reports the national and regional rates of chronic diseases and all-cause mortality in Saudi Arabia, depending on health survey data in 2018. The paper is not suitable for publication in this form; it needs revisions. Some of the modifications are listed below:
Response: We would like to thank the reviewer for feedback on our paper's current state and suggestions for improved revisions. As suggested, we have addressed all suggested modifications listed below.
According to the "Instructions for Authors," the abstract should be a single paragraph and follow a structured abstract style without headings. Please remove the headings.
Response: As suggested, we removed the headings.
The introduction should be significantly improved. The current state of the research field should be included, and related publications should be added. Only general information is given, and international studies about this topic should be reviewed.
Response: As suggested, the introduction section has been improved by including the current state of the research and related publications and giving general global information on pages 1 and 2, lines 43-48 and 40-69, respectively.
Studies about the international status of chronic diseases should be included to discuss the results. The conclusions should consist of key findings and be more specific. Please revise and include some key findings.
Response: As suggested, the international status of chronic diseases has been discussed, and revised the conclusion accordingly on pages 9, lines 242-253 and 297-309; pages 10, lines 324-329.

Reviewer 3 Report
This manuscript looked at event rates for chronic disease and all-cause mortality. The introduction is lack of important details. Formulas in the methods are redundant and analysis methods are sometimes wrong. For example, when you look at mortality rate within women, the denominator should be women only. Results are presented in a way hard to see, for example Figure. Figure 2 is wrong.
Author Response
Reviewer #3
This manuscript looked at event rates for chronic disease and all-cause mortality.
Response: Thank you to the reviewer for constructive feedback on our paper.
The introduction lacks important details.
Response: The introduction has been improved and updated on page 1, lines 43-48; page 2, lines 49-61; 69-70; 74-76; and 77.
Formulas in the methods are redundant, and analysis methods sometimes must be corrected. For example, when you look at the mortality rate among women, the denominator should be women only.
Response: We appreciate the reviewer for these important comments. As suggested, we have addressed formulas in the methods on page 3, lines 115-131.
Results are presented in a way that could be clearer, for example, in Figure. Figure 2 is wrong.
Response: As suggested, the results are presented clearly. Figure 2 has been removed.

Round 2
Reviewer 1 Report
Thank you for the modifications in the draft. All major queries have been answered. I don't find any rational on why authors divided population data into two sub-population (citizens vs residents) but if they have any opinion about it please add that in the draft. If it's just how the data were available and they want to describe the data as it is, that is fine too.
Reviewer 2 Report
It is acceptable in the present form.
Reviewer 3 Report
Unfortunately the main statistical problem is still not resolved. The main finding "The all-cause mortality rate across all causes was 286 per 100,000 people; mostly, with men having a higher rate than women (177 versus 109 per 100,000 people). " is simply not correct. If the calculation is correct, I would expect the mortality rate be higher for one gender, and lower for another, compare to the overall population. Based on current numbers, I believe the mortality calculation needs to be corrected.
Figure 3 is hard to read. Please revise to show proportion in the stacked bar chart, and show number of patients in table below the figure.